# Intensive Intervention on Smoking Cessation in Patients Undergoing Elective Surgery: The Role of Family Physicians

**DOI:** 10.3390/medicina60060965

**Published:** 2024-06-11

**Authors:** Anto Domić, Nataša Pilipović-Broćeta, Milkica Grabež, Nevena Divac, Rajko Igić, Ranko Škrbić

**Affiliations:** 1Primary Health Care Centre, 76100 Brčko, Bosnia and Herzegovina; anto.domic67@gmail.com; 2Department of Family Medicine, Faculty of Medicine, University of Banja Luka, 78000 Banja Luka, The Republic of Srpska, Bosnia and Herzegovina; natasa.pilipovic-broceta@med.unibl.org; 3Department of Hygiene, Faculty of Medicine, University of Banja Luka, 78000 Banja Luka, The Republic of Srpska, Bosnia and Herzegovina; milkica.grabez@med.unibl.org; 4Department of Pharmacology, Clinical Pharmacology and Toxicology, Faculty of Medicine, University of Belgrade, 11000 Belgrade, Serbia; nevena.divac@med.bg.ac.rs; 5The Academy of Sciences and Arts of The Republic of Srpska, 78000 Banja Luka, Bosnia and Herzegovina; igicrajko@gmail.com; 6Department of Pharmacology, Toxicology and Clinical Pharmacology, Faculty of Medicine, University of Banja Luka, 78000 Banja Luka, The Republic of Srpska, Bosnia and Herzegovina

**Keywords:** smoking cessation, intensive intervention, FTND, CO test, cotinine test, family physician

## Abstract

*Background and Objectives*: The aim of this study was to determine the role of physicians in the intensive intervention and education regarding the smoking cessation of patients undergoing elective surgery under general anaesthesia. *Materials and Methods*: A randomised prospective study was conducted in family physicians’ clinics in which smokers of both sexes, aged 21–65 years, without cognitive impairments, and who were not addicted to psychoactive substances voluntarily participated. Four weeks preoperatively, 120 smokers were randomised into two equal groups; the intervention group (IG) underwent an intervention for the purpose of smoking cessation and the control group (CG) underwent no intervention. Biochemical tests were performed in order to determine the smoking status of the participants in the phase of randomisation, one week preoperatively, as well as 40, 120, and 180 days and 12 months postoperatively. The examinees of the IG talked to the physician five times and received 140 telephone messages, leaflets, and motivational letters along with the pharmacotherapy, while the participants in the CG received little or no advice on smoking cessation. *Results*: The results of this study confirmed a significant influence of the intervention and education on the smoking abstinence in the IG compared to the CG (*p* < 0.001). The smokers in the IG had 7.31 (95% CI: 2.32–23.04) times greater odds of abstinence upon the 12-month follow-up than the smokers in the CG. The smokers in the IG who did not stop smoking had a lower degree of dependence and smoked fewer cigarettes (*p* < 0.0001) compared to those in the CG, as well as a multiple times higher prevalence of short- and long-term abstinence. *Conclusions*: It can be concluded that the intensive intervention and education can motivate patients preparing for elective surgery to stop smoking in the short- and long term.

## 1. Introduction

The World Health Organization (WHO) has determined that combustible tobacco smoking is a global problem that has pandemic characteristics, with serious consequences for public health. There are over 1.3 billion smokers in the world, 80% of whom live in societies with reduced socioeconomic standing. Annually, over 8 million people die from active and passive smoking, which is devastating for developing countries [1]. The high morbidity and mortality, particularly in the reproductive population, have forced public authorities to make smoking prevention a key public health priority. Smokers mostly use combustible tobacco products (cigarettes, rarely pipes or cigars, and still much less often e-cigarettes), although the use of e-cigarettes is increasing. [2]. The anti-smoking campaigns conducted among adults in the United States have confirmed that 68% of the adult smokers show an interest in quitting, but only 6.5% to 8.3% of them succeed [3]. According to the report of the World Bank, 41.1% of the adults in Bosnia and Herzegovina (BiH) use various tobacco products; only 12.5% of the smokers have been ready to quit smoking, while only 2% succeed [4]. The reason for such a low rate of success in quitting smoking lies in the highly addictive capacity of nicotine, so smokers, without the help of medical experts, are not ready to embark on a very difficult and uncertain process of abstinence [5]. 

Smokers have six times more frequent lung complications during and after surgery than non-smokers, such as bronchospasm, laryngospasm, and increased mucus production. The wound healing and recovery from surgery are longer and slower in smokers than in non-smokers, which is the reason why they have to stay longer in the hospital [6]. For these reasons, it is important to find sustainable, financially acceptable, and effective ways to quit smoking. There are few studies that have been performed on the impact of an intensive intervention carried out by a physician in order to convince smokers to quit smoking, at least for the short time of the perioperative period [7]. The planned surgical intervention serves as a motivating factor for smoking cessation, and the implementation of a smoking cessation intervention (SCI) during the preoperative period is economically justified and demonstrates a favourable cost-effectiveness ratio for reducing the risk of postoperative complications [8]. However, the optimal duration of the smoking abstinence prior to surgery remains uncertain. Landström et al. suggest that a perioperative smoking cessation four weeks before and after the surgery would lower the incidence of postoperative complications [9], whereas Mills recommends a duration of at least 4 to 8 weeks preoperatively [10]. According to the WHO, extending the smoking abstinence beyond 4 weeks before surgery reduces the postoperative morbidity risk by 19% per additional week [11]. Additionally, Musallam reported that smoking cessation at least one year before a major surgery reduces the risk of postoperative mortality [12]. The evidence from a systematic review and meta-analysis of 38 randomised controlled trials suggests that perioperative SCIs lead to an increase in abstinence rates for up to 12 months following the surgical procedure [13]. The current evidence-based interventions for smoking cessation encompass behavioural treatments, quit lines, web-based support services, and pharmacotherapy. These interventions have proven to be effective in aiding smoking cessation when employed individually, and even greater success has been observed when combining pharmacotherapy with behavioural support [14]. A combination of behavioural therapy and biochemical validation with the help of well-trained medical professionals produced promising results [15]. Tests of the cotinine concentration in urine and carbon monoxide in the exhaled air have demonstrated positive influences on smokers’ efficacy regarding quitting smoking [16].

The aim of this study was to determine the role of physicians in the intensive smoking cessation interventions (ISCIs) of the patients undergoing elective surgery under general anaesthesia. 

## 2. Materials and Methods

### 2.1. Study Participants 

This prospective, randomised, interventional study focused on smoking cessation in patients undergoing preoperative preparation for elective surgery under general anaesthesia. Participants were recruited from the family medicine outpatient clinic in Brčko District of Bosnia and Herzegovina. The study lasted from April 2019 to March 2022, with a follow-up period of 12 months post-surgery. The inclusion criteria comprised smokers of both genders, aged 21–65 years, scheduled for elective surgery, with at least low nicotine dependence confirmed by the Fagerstrom questionnaire and biochemical validation tests. Surgical procedures included hip, knee, inguinal hernia, or gallbladder surgery. Exclusion criteria included current pregnancy and breastfeeding, severe chronic or advanced diseases, psychiatric illness, drug or alcohol abuse, and any form of cognitive impairment.

### 2.2. Ethical Considerations

Participants were informed of the study’s purpose, study design and protocol, associated risks and benefits, as well as the study’s duration. The patients were advised to consult their family members or physician before signing the consent for participation, which was voluntary and with the notion they could leave the study at any time. All respondents who expressed their interest to participate in the study and who met the inclusion criteria provided signed informed consent.

### 2.3. Study Design

A total of 120 participants were randomly assigned to two equal parallel groups, the intervention group (IG) and the control group (CG), by a family physician using a random number generator. The respondents signed the “Declaration on temporary cessation of smoking” and filled out the General Health Questionnaire, the Fagerstrom Test for Nicotine Dependence (FTND) [17], Hollingshead’s Questionnaire [18], and the declaration of acceptance of communication, while the researcher signed the declaration of data confidentiality. In the IG (N = 59), participants underwent an intensive intervention with pharmacological support, while, in the CG (N = 61), participants received only advice on smoking cessation, which is typically provided to all patients during pre-surgery medical examinations. 

A direct seven-month intensive intervention involved activities like determining the smoking status using the Fagerstrom test and biochemical validation tests, a face-to-face interview with the respondent that was organised through five planned test phases, or during a visit to the family physician at regular check-ups due to underlying disease. Face-to-face interviews with motivated respondents to quit smoking were conducted according to the behavioural “5A” method (Ask, Advise, Assess, Assist, and Arrange). Respondents who lacked motivation were applied the 5R model (Relevance, Risks, Rewards, Roadblocks, and Repetition). The process of direct intervention was carried out through 6 phases (F): F-0: Randomisation phase—four weeks before surgery;F-1: From three to seven days before surgery;F-2: Forty days after surgery;F-3: Four months after surgery;F-4: Six months after surgery;F-5: Follow-up was conducted 12 months after surgery.

Indirect intervention means sending short messages using SMS, Viber, WhatsApp, sending leaflets, letters, and other suitable material via e-mail. During the investigation, the respondents in the IG received 140 SMS messages; in the perioperative period (30 days before and 40 days after surgery), they received messages every day, and after that every other day until six months postoperatively. A nurse from the family physician’s team was responsible for sending messages in a timely manner. The principal investigator spoke weekly by telephone with each respondent during the preoperative period and four months postoperatively, and twice a month in the period from the fourth to the sixth month postoperatively. The contents of SMS leaflets, brochures, and personal letters of encouragement that were sent to patients during the period of intensive intervention were unified. Respondents of IG were recommended to take bupropion or nicotine substitutes such as nicotine-containing chewing gums or nicotine patches. Bupropion therapy lasted three months, and it started a week before quitting smoking. In case of side effects such as depressed mood, agitation, suicidal thoughts, or suicidal behaviour, the respondents were advised to stop taking the drug.

### 2.4. Measurements

Self-assessment of smoking status and nicotine addiction was determined by means of Fagerstrom’s questionnaire, and honesty was checked with biochemical validation tests: urine cotinine test and carbon monoxide concentration test in exhaled air. The presence of cotinine in urine was determined using the qualitative “Accutest NicAlert Urine Cotinine Test” (Jant Pharmacal Corp. Encino, CA, USA). The concentration of carbon monoxide in exhaled air was determined using “The Micro CO meter rates” (Bedfont Instruments, Kent, UK). The carbon monoxide concentration value was expressed in ppm (parts per million) and ranges from <2 for non-smokers who were not exposed to passive smoking, ˃3<6 ppm in passive smokers, moderate smokers ˃7<9 ppm, and >10 ppm in passionate smokers. The successful abstinence was defined if the Fagerstrom score was 0, if no cotinine was detected in urine, and if the concentration of carbon monoxide in the exhaled air was less than 2 ppm. Participants who did not respond to the physician’s call or gave up on the study during the program or during the 12-month follow-up period were treated as smokers when presenting the results.

### 2.5. Statistical Analysis

We determined the sample based on the range presented in the literature and our assumed goal of achieving preoperative abstinence of 5% in the CG and 25% in the IC. In relation to a three-year average of 650 surgical operations under general anaesthesia in Brčko General Hospital, we determined a sample of 120 smokers that provided 95% confidence to establish an absolute difference in abstinence of 25% at the two-tail Fisher’s exact test level. For numerical variables, the arithmetic mean and SD were used, while the categorical variables were compared by chi-squared test. For the differences between numerical variables in test phases, the Friedman test was used. The Cohen’s kappa coefficient was used to assess the level of compliance of patients’ statements about abstinence. The influence of sociodemographic factors on a patient’s decision to abstain was evaluated by a univariate and multiple logistic regression model and presented as an odds ratio (OR) with a 95% confidence interval (CI). The relative risk (RR) of smokers not abstaining after the intervention was estimated by calculating the risk ratio and 95% CI. Statistical significance was confirmed at *p* < 0.05. For data processing, the statistical program SPSS 25 (SPSS Inc., Chicago, IL, USA) was used.

## 3. Results

At the beginning of the study, 183 patients were interviewed, and, according to the criteria, 120 (65.6%) patients were included in the study. One IG respondent did not provide data on marital status and education, so further calculations for these variables were performed with 58 respondents. The long-term effect of the intensive intervention was checked 12 months after the end of the program, and 74.5% (44/59) of the respondents from the IG responded to the call, while 48/61 (78.6%) of the respondents from the CG were available to take the Fagerstrom test and undergo the biochemical validation of the results. The respondents of both groups who did not respond to the call were considered as smokers. Among the respondents of both groups, there were no significant differences in any sociodemographic or smoking characteristics (*p* > 0.005). The basic sociodemographic characteristics and smoking statuses of the study participants are shown in Table 1.

All the respondents in the randomisation phase were smokers, but, after the initial intervention, the number of abstainers in the IG increased by 60%, and the same was recorded 40 days postoperatively; four months postoperatively, half of the respondents abstained, and, at the six-month postoperative check-up, there were slightly more than 42% abstainers. A one-year follow-up showed that a third of the respondents abstained. In this group, the number of abstainers differed significantly during all the phases of the study (*p* = 0.002; Table 2; Figure 1).

A significant number of the respondents in the CG also abstained before the operation (11%) and during the four-month postoperative period. However, at the end of the study, abstinence was determined only in every twentieth respondent, and 27 participants did not respond to the doctor’s call after the 12-month follow-up period and were treated as smokers. The change in the number of abstainers in the CG from seven days before surgery until the one-year follow-up had no statistical significance (*p* = 0.212). 

The smoking status of the respondents of both groups in the phase of randomisation was the same, (*p* = 1), but, in all the other phases, the number of abstainers in the IG was several times higher than in the CG (*p* < 0.001). Complete compliance with the statement regarding abstinence, which was confirmed via biochemical validation, was noticed only in the randomisation phase (*p* = 0.986), and, in the other cases, the majority of the participants claimed that they did not smoke, but the biochemical validation showed the opposite. Regardless of these discrepancies, the concordance of the participants’ statements and biochemical validation results was very high (*p* < 0.001): the kappa coefficient ranged from 0.793 to 0.906 (Table 2).

The Fagerstrom score between the respondents of the IG and those from the CG was similar only in the randomisation phase (*p* = 0.484). Just before the surgery, the average value of the Fagerstrom score in F-1 was significantly lower in the IG (*p* < 0.0001). Until the end of the study, the average Fagerstrom value in both groups did not differ significantly (pFIG(4) = 0.068 and pFCG(4) = 0.064, but its value in the IG was always several times lower compared to the CG (*p* < 0.001; Table 3).

Based on the FNDT value, the degree of nicotine addiction of the respondents was similar in both groups in the randomisation phase (*p* = 0.984), and 80% of the participants had Fagerstrom scores > 4. A significant difference between the groups was observed after the first intervention when a sevenfold decrease in the number of smokers with medium, moderate, and severe addiction was recorded in the IG compared to the CG. Comparing the degree of dependence between the first and fourth stages of the test, it significantly decreased only in the IG (pFIG(2–5) = 0.007) but not in the smokers from the CG (pFCG(2–5) = 0.123) (Table 3).

At the beginning of the study, the difference in the number of cigarettes smoked per day was not statistically significant (*p* = 0.574). The first change in the number of smoked cigarettes was recorded before the operation, when the respondents of the IG reduced the number of cigarettes smoked per day by an average of three times, while those in the CG decreased it by one fifth (*p* < 0.0001. During the six-month postoperative period and the one-year follow-up period, the respondents of the IG smoked significantly less compared to the CG respondents (*p* < 0.001 Table 3, Figure 1).

The pharmacotherapeutic support with bupropion was accepted by 22% (13/59) of the respondents of the IG, and 8.5% (5/59) of them took the medication according to the protocol. According to the protocol, 9/59 (15.3%) of the respondents of the IG used nicotine replacement therapy, but the effect on abstinence was not statistically significant (*p* = 0.20).

The impact of the intervention methods and sociodemographic factors using univariate and multiple logistic regression models was also analysed. Using the univariate model, only belonging to the IG was significant, but not sex, age, marital status, education, and socioeconomic status. The multiple model analysis showed that belonging to the IG and having a high socioeconomic status were significant (OR: 10.92: 95% CI: 6.93–17.2) (OR: 4.77 (95% CI: 1.17–19.56), respectively. The respondents of the CG have a higher risk of being a smoker (RR: 5.17 (95% IP: 1.88–14.22) than the respondents of the IG (RR: 1.41 (95% IP: 1.33–2.24).

## 4. Discussion

Our study was conducted in primary health care by a family medical team (family physician and nurse) among surgical patients as this setting is seldom investigated for preoperative smoking cessation intervention, and therefore our study is among the few that have presented short-term and long-term positive outcomes. Good results were achieved by applying an intensive intervention for a period of seven months with a combination of direct and indirect measures and pharmacological support with periodic checking of smoking status using the Fagerstrom test and biochemical validation. Before surgery and 40 days postoperatively, the motivation for abstinence was higher, with 60% of the IG respondents quitting, while half of the respondents four months postoperatively and four out of ten were non-smokers six months after surgery. 

Although this smoking cessation method is available in the literature, such studies are rare, as confirmed by Berlin et al. [19], who listed only eight randomised controlled studies that met the criteria of an intensive intervention, of which only three revealed a statistically significant relationship (connection) between the preoperative interventions and a 12-month period of smoking cessation. They stated that the one-year abstinence rate ranged from 13% [20] to 33% [21] in the IG versus 4.4% [22] to 19.7% [19] in the CG. In a Cochrane database systematic review, Thomsen et al. [23] showed the evidence for the successful short-term abstinence in patients who underwent a preoperative intervention, while the effects on the long-term abstinence remain unclear [24]. A more recent study conducted in the Netherlands has shown that an intensive intervention applied daily by a family physician produced long-term effects in 28.3% of the smokers [25,26,27]. According to our experiences from this research, primary health care provides greater chances for short-term and long-term success due to the fact that patients visit their family physicians for various reasons and health needs. The physician is familiar with the health and social circumstances of the smoker’s family. In addition, it is necessary for a family physician to take advantage of every patient visit and apply a comprehensive approach related to an intensive smoking cessation intervention [28,29,30]. What negatively affects the success of this method is the knowledge of the high prevalence of smoking among young people, students, and physicians, and that smoking is a socially acceptable norm [31,32,33,34]. In order to raise awareness regarding the harmful effects of smoking, Igic and Bernaciak [35] announced the initiative to make the smoking status of candidates eliminatory when enrolling in a medical faculty, but this initiative has not yet met with fruitful results [34,36]. This social attitude towards smoking has a negative effect on the motivation for active participation in a smoking cessation program. In the literature, the following have been mentioned among the most important motivating factors: health, economic, financial, and social factors [37]. The strongest motivating factor for quitting smoking is health, so we used the upcoming elective surgery as a trigger for the abstinence that the smoker was supposed to undergo [38].

Active participation in the smoking cessation program begins with the signing of the informed consent, in which the respondent receives information about the harmful effects of smoking as well as the health benefits during and after surgery. When making a decision to quit smoking, it has been shown to be important to consult with family, friends, and the physician regarding that decision. A smoking cessation contract signed with a physician additionally strengthens self-confidence because it seems that the contract serves as a token of obligation, which has a positive psychological effect on the execution of the decision [39]. Such experiences have been described in the literature, but there is currently not enough scientific evidence for routines to be applied [40]

Self-assessment tests of smoking status, the FTND, and biochemical validation [41] help to make smoking cessation successful [42]. Regardless of the fact that the FTND is based on the self-report of the subjects, in combination with biochemical validation, it represents a sufficiently powerful tool that provides reliable data on smoking status. These tests are not only informative but also therapeutic because, even after the first self-assessment, the CG subjects had lower FTND values on average, which is why these values in the IG were several times lower on average. Biochemical validation tests complement the results of the FTND test because the value of CO in exhaled air is highest just after smoking, while cotinine remains in the body longer, so, in this way, we monitor the usefulness of the self-assessment test. Objectifying the concentration of carbon monoxide in the exhaled air also encourages the smoker to “leave cigarettes” [43]. Herbeć and associates published a study in which they stated the importance of presenting the results of the carbon monoxide concentration in exhaled air as a highly motivating incentive to quit smoking [44]. We found a similar reaction in our study, where the respondents actively participated in measuring the concentration of carbon monoxide and received information about the harmfulness of this gas to health with more attention than when it came to some other chemicals from cigarettes [45]. Such a reaction is also caused by determining the presence of cotinine in urine; however, during the biochemical validation, the respondents paid more attention to and showed more interest in the measurement of the carbon monoxide concentration in exhaled air than to the process of measuring the cotinine in urine. We assume that the reason is that it is a qualitative method whose measurement results are not as exact as in carbon monoxide measurement [39,46].

Telephone and face-to-face conversations provide the respondents the opportunity to dispense with the myth about the impossibility of quitting smoking, relieve obsessive thoughts, and overcome the desire for a cigarette. Physicians must take care not to increase patients’ existing fear of surgery with advice and behaviour because it is known that anxiety increases the desire for cigarettes [47,48]. In such circumstances, the physician, with his/her expertise and behaviour, must be a constant support and authority to the former smoker in order to lead him/her out of the labyrinth of smoking addiction [49]. In this study, it was shown that the smokers were much more motivated to quit smoking and achieved long-term abstinence if advised by a non-smoking physician.

The structured conversation according to the “5A” behavioural method and the “5R” method was applied to those who lacked motivation. These techniques enable smokers to identify their personal frustrations, discover smoking triggers, and develop alternative ways to avoid smoking. They received support from professionals for the joint preparation of “your day without smoke”, such as advice for overcoming the craving for a cigarette, a physical activity plan, organizing free time, and adjusting their diet menu, all with the aim of training them to successfully avoid relapse [50]. Short messages sent from a physician’s office via a digital platform have been demonstrated as a useful tool to keep the abstainer “awake” and motivated to overcome the crisis and remain a non-smoker. In their review articles, Scott-Sheldon [51] and Wittaker [52] claimed that a text message intervention with smoking cessation is practical, cheap, and provides unequivocal help in the process of quitting smoking,

Pharmacological support with bupropion and nicotine replacement therapy (NRT) contributed to positive attitudes for ex-smokers to overcome the abstinence crisis more easily [53,54], and they were recorded as long-term abstainers. NRTs were chewed by several respondents, although the majority of the patients did not accept the NRT protocol. In this study, due to side effects, most of the respondents were convinced that, in the process of quitting smoking, bupropion would harm their health more, and they preferred not to take it.

The sex of the respondents has no significance regarding this method of smoking abstinence, so it seems that both men and women are equally prone to smoking and abstinence [55]. Different smoking motives of women and men are listed in the literature, so it is stated that women smoke more often in situations when they are under stress and other emotional states, while men smoke for social reasons. Intensive interventions have an individualised approach, which enables both women and men to start and succeed in the process of quitting smoking. Women need support and positive motivation in that process, and both sexes expressed their desire for help without judgment or criticism. Unlike with men, in women, the reward and punishment system has no effect on smoking status [56].

SES has a long-term significant influence on smoking cessation, and a multivariate analysis showed that smokers with a high SES quit smoking five times more often than smokers with a low SES. In the available literature, it is indicated that socially vulnerable groups have the highest frequency of smoking and a higher level of addiction. Such persons have low social support in their intention to quit smoking; therefore, these smokers develop an indifferent attitude towards the need to quit smoking [57,58].

### Limitations of This Study

The sample was determined on the basis of a three-year average number of operations under general anaesthesia and not on the basis of the entire population of smokers, so the sample size may be questionable. Only smokers who were preparing for elective surgery were included in the research, so the success of the method on the general population of smokers is unclear, especially healthy smokers, adolescents, and those suffering from some chronic disease. The respondents who dropped out of the study were still considered as smokers even though the researcher did not know their smoking status. However, teamwork between the family physician, surgeons, anaesthesiologists, and other health professionals would greatly improve this method and make it easier for smokers to make the decision to quit smoking and live without cigarettes. 

## 5. Conclusions

Family medical teams are the best option for the short- and long-term implementation of smoking cessation programs for patients preparing for elective surgery under general anaesthesia. The comprehensive approach of applying an intensive intervention helped 59% of the respondents in the IG to stop smoking in the short term (up to 40 days postoperatively), which is five times more (11.5%) than in the CG. In the long term, the number of IG non-smokers decreased, but they were still 42% after six months postoperatively, while, in the CG, they were less than 5%. One third of the IG respondents were still non-smokers one year after the program, while the corresponding value was 8% in the CG. If the synergistic action of different services of the health system (family physician, surgeon, anaesthesiologist, and other specialists) were achieved, long-term beneficial results would be achieved in a shorter time. Determining the smoking status and advising on the harmful effects of smoking and methods for quitting must be a daily routine activity of family physicians.

## Figures and Tables

**Figure 1 medicina-60-00965-f001:**
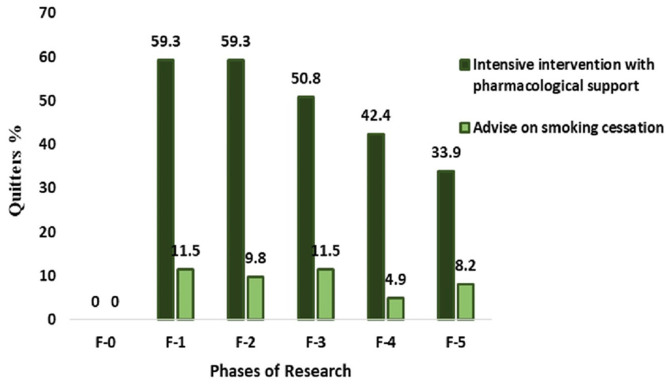
Number of quitters by FTND according to research phases. FTND—Fagerstrom Test for Nicotine Dependence.

**Table 1 medicina-60-00965-t001:** Sociodemographic characteristics of study participants.

Variables		Total N = 120	IG N = 59	CG N = 61	*p*
Sex, n (%)					
	Male	62 (51.7)	33 (55.9)	29 (47.5)	0.357
	Female	58 (48.3)	26 (44.1)	32 (52.5)	
Age (years), mean (SD)		48.2 (11.8) 21–65	46.9 (11.9) 21–65	49.4 (11.7) 25–65	
Marital status, n (%)					
	Married	93/119 (78.2)	47/58 (81.0)	46 (75.4)	0.457
	Single	25/119 (21.0)	11/58 (19.0)	14 (23.0)	
	Widower/Widow	1/119 (0.8)	0/58 (0)	1 (1.6)	
Education, n (%)					
	Elementary school	10/119 (8.4)	5/58 (8.6)	5 (8.2)	
	Trade	11/119 (9.2)	7/58 (12.1)	4 (6.6)	
	Secondary school	59/119 (49.6)	28/58 (48.3)	31 (50.8)	
	School of higher education	12/119 (10.1)	5/58 (8.6)	7 (11.5)	0.905
	College	24/119 (20.2)	12/58 (20.7)	12 (19.7)	
	Postgraduate education	3/119 (2.5)	1/58 (1.7)	2 (3.3)	
SES, points		34.2 (11.6) 11–63	33.8 (10.5) 17–62	34.6 (12.6) 11–63	
SES, n (%)					
	Low (8–27)	44/119 (37.0)	21/58 (36.2)	23 (37.7)	
	Medium (28–47)	60/119 (50.4)	30/58 (51.7)	30 (49.2)	0.959
	High (48–66)	15/119 (12.6)	7/58 (12.1)	8 (13.1)	

IG = patients undergoing intensive smoking cessation intervention; CG = patients subjected to usual smoking cessation advice; SD = standard deviation; SES = socioeconomic status.

**Table 2 medicina-60-00965-t002:** The results of the subjects’ abstinence were determined by self-assessment of smoking status and biochemical validation.

Phases of Intervention			IG	CG	RR (95% CI)	*p*
			n = 59	n = 61		
Smoking status 7 days before surgery						
	Abstinence FTND n (%)					
		Quitters	35 (59.3)	7 (11.5)	5.17 (2.49–10.7)	0.0001
	Abstinence biochemically validated * n (%)					
		Yes	29 (49.2)	5 (8.2)	5.99 (2.48–14.44)	0.0001
Smoking status 40 days postoperatively						
	Abstinence FTND n (%)					
		Quitters	35 (59.3)	6 (9.8)	6.03 (2.74–12.27)	0.0001
	Abstinence biochemically validated * n (%)					
		Yes	32 (54.2)	4 (6.6)	8.27 (3.11–21.94)	0.0001
Smoking status 4 months postoperatively						
	Abstinence FTND n (%)					
		Quitters	30 (50.8)	7 (11.5)	4.43 (2.11–9.29)	0.0001
	Abstinence biochemically validated *					
		Yes	29 (49.2)	5 (8.2)	6.89 (2.90–16.37)	0.0001
Smoking status 6 months postoperatively						
	Abstinence FTND n (%)					
		Quitters	25 (42.4)	3 (4.9)	8.61 (2.74–27.01)	0.0002
	Abstinence biochemically validated *					
		Yes	24 (40.7)	4 (6.6)	6.20 (2.29–16.79)	0.0003
Follow-up after 1 year						
	Abstinence FTND n (%)					
		Quitters	20 (33.9)	5 (8.2)	4.13 (1.66–10.29)	0.0023
	Abstinence biochemically validated *					
		Yes	18 (30.5)	3 (4.9)	6.20 (1.92–19.96)	0.0022
	pCQ(6) abstinence according FTND				<0.001	0.002
	pCQ(5) abstinence according to FTND				0.037	0.212
	pCQ(6) * abstinence biochemically validated				<0.001	0.06
	pCQ(5) * abstinence biochemically validated				0.287	0.896

Do not smoke: FTND test = 0 points; smoke FTND test > 0; * validated abstinence: yes = concentration of CO in exhaled air ≤ 2 ppm + absence of cotinine in urine; no = other; IG = patients undergoing intensive smoking cessation intervention; CG = patients subjected to usual smoking cessation advice; *p* = *p*-value for testing the difference in the frequency of abstainers between IG and CG, for each study phase separately, chi-squared test; pCQ(6) = *p*-value for testing the difference in the frequency of abstainers within a group among all six study phases (from first contact including follow-up a year after), Cochran Q test; pCQ(5) = *p*-value for testing the difference in the frequency of abstainers within the group among the five study phases (from 7 days before OP including follow-up a year after), Cochran Q test; RR—relative risk; FTND—Fagerstrom Test for Nicotine Dependence.

**Table 3 medicina-60-00965-t003:** Degree of dependence according to Fagerstrom, and number of cigarettes smoked per day in different study phases.

Study Phase	F-0		F-1		F-2		F-3		F-4		F-5 (Follow-Up)		*p*-Value	
Group	IG	CG	IG	CG	IG	CG	IG	CG	IG	CG	IG	CG		
FTND—points														
*p*-value	*p* = 0.484		*p* < 0.001		*p* < 0.001		*p* < 0.001		*p* < 0.001				p_FIG(5)_ < 0.001 p_FCG(5)_ < 0.001 p_FIG(4)_ = 0.068 p_FCG(4)_ = 0.064	
Mean (SD)	5.58 (2.31)	5.26 (2.24)	1.29 (2.22)	4.10 (2.56)	1.00 (2.01)	3.95 (2.48)	1.10 (2.00)	3.86 (2.43)	1.27 (1.87)	4.07 (2.21)	2.0 (2.9)	5.4 (2.1)		
>4 n%	48 (81.4)	48 (78.6)	7 (11.9)	36 (59.0)	5 (8.5)	31 (50.8)	5 (8.5)	32 (52.5)	5 (8.5)	33 (54.1)	22 (37.3)	43 (70.5)	p_FIG(6)_ < 0.001 p_FCG(6)_ < 0.808 p_FIG(5)_ = 0.527 p_FCG(5)_ = 0.426	
<4 n;%	11 (18.6)	13 (21.3)	52 (88.1)	23 (37.7)	49 (83.1)	25 (40.9)	48 (81.4)	24 (39.4)	47 (79.6)	23 (37.7)	22 (37.3)	5 (8.2)		
Nicotine dependence n (%)														
*p*-value	*p* = 0.984		*p* < 0.001		*p* < 0.001		*p* < 0.001		*p* < 0.001		*p* < 0.001			
High (8+ points)	11 (18.6)	10 (16.4)	2 (3.4)	5 (8.5)	1 (1.9)	5 (8.9)	1 (1.9)	4 (7.1)	1 (1.9)	3 (5.4)	2 (3.4)	2 (3.27)	p_FIG(6)_ < 0.001 p_FCG(6)_ < 0.001 p_FIG(5)_ = 0.007 p_FCG(5)_ = 0.123	
Medium (5–7)	28 (47.5)	29 (47.5)	3 (5.2)	21 (35.6)	3 (5.7)	17 (30.4)	3 (5.8)	18 (32.1)	3 (5.8)	20 (35.7)	14 (23.7)	33 (54.1)		
Moderate (3–4)	15 (25.4)	16 (26.2)	5 (8.6)	18 (30.5)	3 (5.7)	19 (33.9)	3 (5.8)	19 (33.9)	4 (7.7)	24 (42.9)	8 (13.5)	8 (13.1)		
Low (1–2)	5 (8.5)	6 (9.8)	13 (22.4)	8 (13.6)	11 (20.8)	9 (16.1)	15 (28.8)	8 (14.3)	19 (36.5)	6 (10.7)	2 (3.4)	2 (3.3)		
None (0)	0 (0)	0 (0)	35 (60.3)	7 (11.9)	35 (66.0)	6 (10.7)	30 (57.7)	7 (12.5)	25 (48.1)	3 (5.4)	20 (33.9)	4 (6.5)		
Number of cigarettes smoked per day														
Mean (SD)	17.8 (7.2)	17.0 (8.3)	5.6 (8.2)	13.3 (8.3)	4.4 (7.5)	13.0 (8.1)	5.1 (7.5)	13.0 (8.4)	5.6 (6.6)	14.2 (7.6)	8.8 (8.7)	15.3 (6.4)		
*p*	0.574		0.0001		0.0001		0.0001		0.0001		0.0001			

Numerical variables are presented as whole numbers and (percentages); IG—patients undergoing intensive intervention to quit smoking; CG—patients subjected to usual smoking cessation advice; FTND—Fagerstrom Test for Nicotine Dependence; F-0 = smoking status one month before surgery; F-1 = smoking status 3–7 days before surgery; F-2 = smoking status 40 days after surgery; F-3 = smoking status 120 days after surgery; F-4 = smoking status 180 days after surgery; F-5 long-term follow-up; OP = operation; *p* = *p*-value for the comparison between IG and CG for each study phase, chi-squared test for Fagerstrom score—level of dependence; p_FIG(6)_ = *p*-value for testing the difference in the frequency of abstainers from IG among all five study phases (from first contact to 6 months after OP), Friedman test; p_FCG(6)_ = *p*-value for testing the difference in the frequency of abstainers from CG among all five study phases (from first contact to 6 months after OP), Friedman test; p_FIG(5)_ = *p*-value for testing the difference in the frequency of abstainers from IG among the four study phases (from 7 days before OP to 6 months after OP), Friedman test; p_FCG(5)_ = *p*-value for testing the difference in the frequency of abstainers from CG among the four study phases (from 7 days before OP to 6 months after OP), Friedman test.

## Data Availability

The datasets used and/or analysed during the current study are available from the corresponding author upon reasonable request.

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
