# Peer review of "Intensive Intervention on Smoking Cessation in Patients Undergoing Elective Surgery: The Role of Family Physicians"

_medicina, 2024, doi:10.3390/medicina60060965_

Round 1
Reviewer 1 Report
Comments and Suggestions for Authors
The topic of the manuscript is of great interest for primary healthcare and for public health, as it tackles one of the main preventable causes of disease.
Overall I suggest a rewriting and a a more structured presentation of the methods and results section.
Abstract - the abstract contains contradictory data compared to the methods and results section (for example - line 21- In the abstract "aged 18-56", in methods "18-65 year" and in the results section, table 1 -21-65 years old. Please double check the numbers and correct them.
Line 27-29 "Th examinees in Group A talked to the physician five times and received 120 telephone....." while in the methods section, lines 131-137 "respondents received 170 SMS messages...the principal investigator spoke weekly...." the information from the abstract seems to contradict the information from the methods part of the manuscript. check the protocols you had and implemented and correct the data accordingly.
Introduction
Line 50 "but only 3-6% of them succeed" - add reference
The introduction needs more context on the methods used for initiating quit behaviour and methods for consolidating this behaviour.
The content between lines 56-63 needs to be connected more to the idea of smoking cessation. Are patients preparing to undergo elective surgery more inclined to quit smoking? Are there relevant studies? Is elective surgery alone a factor for inducing smoking cessation?
Clearly state the objective of the research.
Methods and respondents
The title of the section should be changed. There are no "responders", go with a more classical "Material and Methods".
Overall the methods section needs rewriting for more clarity. I recommend to structure the methods part using subtitles that describe different parts of the study for more clarity (participants, intervention, measurements etc...).
The moment of completing the Fagerstorm score is not clear. On lines 77-78 it is says that "...at least low nicotine dependence was confirmed by the Fagerstorm score.." as an inclusion criterion, but later on, lines 93-94 it says that the score was filled in by the participants after randomization (which implies that they were already included in the study). Please clarify the methods and the steps used for the study. At the beginning of the results section, you describe the method of interviewing. This is better suited for the methods part.
Lines 111-113 what was the reason for considering drop-out patients as being non-smokers? This might influence the final results of the study
How did the intervention was different in F-1 between the two groups?
Statistical analysis - Try to structure more this section, as it elaborates on a significant number of comparisons, and can become unclear.
How was the multiple logistic regression model constructed?
In the methods part you describe 120 patients, but the analysis from Table 1 describes only 119 patients.
Results
The results section offers a lot of details. Therefore, I recommend structuring the whole results section following the different objectives of the conducted analysis. This will add more clarity to the results, which in this form are very difficult to follow and understand.
The results in the tables seem like a "race for p-values". For example, in Table 1, there is a p-value for each line, while this should be calculated only for each category (for example do not compare the proportion of males and the proportion of females separately, but consider them as a group and compare the proportion based on sex. the same goes for Marital status, Education and SES).
For Table 2 it is not clear (or appropriate) to put OR and RR. Also the part with pCQ should be a separate table.
Consider changing Table 2 to two charts (similar to Figure 1. One for declared abstinence and one for "measured" abstinence. This could be more informative than a very cluttered table.
For Table 3 How was Fergurstorm Test conducted on patients that quit smoking?
In Table 3 - Even though, as stated in lines 179-180 48 patients responded to the call at 12 months, in the table adding the numbers leads to 49. Please double-check your numbers for correctness.
Discussion
Lines 303-305 clarify the sentence. Introduce the comparator used.
A point of discussion could be the relapse rate between the two groups (as declared by participants). For Group A the relapse was approximately 43% (15/35) while in group B the relapse was 28% (2/7). Thus, it might be interesting to explore (if found necessary or relevant by the authors) why this large gap in relapse was observed and how this might impact future programs.
Lines 327-330 - For this study, was the level of smoking among the medical professionals participating in the study evaluated? (or patients' knowledge of the smoking status of the investigators?).
Comments on the Quality of English Language
Generally, the language is clear. A check for typos is necessary.
Reviewer 2 Report
Comments and Suggestions for Authors
Thank you for the possibility to peer review this manuscript representing a very important clinical research area. Overall, the design and related conduction constitute a solid basis for writing a brilliant and clinically relevant article. However, the manuscript is very long and contains an overwhelming number of statistical analyses, the inclusion of working tables instead of condensed tables, the repetition in the text of the numbers from the tables. Furthermore, it lacks the necessary ethical considerations for clinical trials and the most important perspectives (patients, clinical, healthcare, societal or research) as well as a logic presentation of the bias, strength, and limitations. All of this counteracts the presentation of a very good clinical trial.
Overall, the English language is very good, except for the use of obsolete and stigmatizing terms like “alcoholics” and “addicts” in L 80. Please, use the updated terms from the newest ICD version or from the WHO reports on similar areas, e.g., persons with alcohol use disorder and persons with addiction. Likewise, the sentence in line 86-87 expresses an obsolete approach about that somebody should be sanctioned if they want to leave the study, while the authors probably mean that it will not impact the treatment of the patients. In L 101 the term honesty should be avoided as that is also stigmatizing, and in L 327 the terms underdeveloped and developing countries could easily be replaced by the less stigmatizing very low- and low-income countries. (Please, be aware that the smoking rate in your country is considered high for a high-income country.)
The introduction
L 56-69: Please be aware that the ref 5 is a protocol for an RCT (not an observational study). Why not refer to a published review? E.g., Yoong SL, Tursan d’Espaignet E, Wiggers J, St Claire S, Mellin-Olsen J, Grady A, Hodder R, Williams C, Fayokun R and Wolfenden L. WHO tobacco knowledge summaries: tobacco and postsurgical outcomes. Geneva: World Health Organization; 2020. License: CC BY-NC-SA 3.0 IGO or other reviews? Please, also be aware of the international recommendations for references; e.g., Wong J, An D, Urman RD, Warner DO, Tønnesen H, Raveendran R, Abdullah HR, Pfeifer K, Maa J, Finegan B, Li E, Webb A, Edwards AF, Preston P, Bentov N, Richman DC, Chung F. Society for Perioperative Assessment and Quality Improvement (SPAQI) Consensus Statement on Perioperative Smoking Cessation. Anesth Analg. 2020 Sep;131(3):955-968. doi: 10.1213/ANE.0000000000004508. Please, introduce the effect of pharmaceutical support in addition to the behavioral therapy and the biochemical validation, as you use that in your study of intensive intervention.
If you want to use abbreviations, then SCI is usually used for the smoking cessation intervention and I-SCI for the intensive programs.
Please, specify your aims and the related hypotheses to help the readers better understand this important study.
Methods and respondents
L 71: An RCT is usually prospectively performed and mentioning it here may confuse the reader.
L 74: Thank you for presenting the study period. Could you please, shortly add the period of recruitment?
L 82: “Respondents … were given informed consent”: Hopefully, the you men that the respondents gave informed consent?
L 86-87: See above regarding stigma.
L 90: Please, do not introduce more terms for the readers to remember, like group A and B. It would help the readers just to use general terms, like intensive intervention and advise or intervention group (IG) and control group (CG).
L 90-92: Please, shortly inform is the family physician in charge of the randomization was involved in the study otherwise or is part of the author group (by initials)?
L 97: Please, shortly inform which pharmaceutical support you offer and if it was offered free of charge or how it was paid? Please, add how many meetings and average time spend on each meeting as this would help the readers to better understand your program.
L 101: See above. Fagerström’s test for nicotine dependency is not developed to measure abstinence, which should be mentioned in the (new) bias, strength, and limitation paragraph.
L 109: Do you mean defined instead of recorded?
L 141+394: Please come and replace this commercial name with chewing gum (if this is correct?)
L 148-151: Please add to the references for choosing the effect. The power calculation is not easy to follow. By re-calculating the power using the chosen incidences, the results are 2 * 27 = 54 (0.8 power) and two times 35 = 70 (0.9 power). In addition, the absolute difference of the chosen incidences is 30%. If you instead have chosen the 25%, which is also mentioned in the text you would reach 2 * 57 = 114, which is close to the number included in the study. It would be helpful for the readers, if you could make the numbers and the text in agreement with each other.
L 152-168: The statistical paragraph needs revision as the number of tests for significance is far too high to be scientifically meaningful. First, you need to decide if you expect the data to be parametric or non-parametric and then choose the related statistics, instead of jumping between the two methods. Today, you very seldom test for differences among the groups at baseline in an RCT (table 1). Bonferroni’s correction is generally used to correct the level of p-value in case of a high number of independent tests, not just for significant tests. Please, choose between analyzing and presenting either the RR or the OR. It may be helpful for the readers if you choose the RR, as that is easier to understand and translate, in general. Please, consider using the area on the curve test for the FTND results and number of cigarettes smoked over time to reduce the number of tests for significance (please, see the results below)
If you want to help future researchers performing a review on this area, you could present the mean and the standard deviation in the supplemental material, in addition to the median and the range used in this article. Then you would need to add this argument in the statistical paragraph.
L 169: please, add a trial profile, like the consort figure.
L 169: please, add a paragraph on your ethical considerations and the ID number for the registration of your protocol.
Results
Overall, please present condensed tables; e.g., in Table 2 please, presence of quitters by FTND and biochemically validated, respectively, in numbers and percentage. You don't need to add the opposite outcomes. Please, also avoid repeating the data from the tables in the text. It seems that this is a study without any missing data which is quite unusual in clinical trials. Therefore, it should be mentioned in there (new) paragraph on bias strength and limitations. If you instead have imputed the worst case scenario according to the Russel’s criteria for reporting SCI in RCTs, then you should of course mention that (and include this in the same paragraph.
Please, consider removing the table 4 to the supplemental material and shortly add the results in the text only; e.g., in the univariate model, only belonging to the IG was significant, but not sex, age, marital status, education and socioeconomic status. The multiple model analysis showed that belonging to the IG and having a high socioeconomic status were significant (OR …. 95% CI …..) and (OR …. 95% CI …..), respectively.
Discussion
Overall, the discussion would benefit from putting the special setting, program, and patient group in focus. Instead, the more general part of the smoking addiction and intervention cessation reduced and thereby leave room for the lacking paragraphs on Perspectives and Bias, strengths, and limitations.
L 302: The very brief summary should include the main results not new hypotheses. Please focus upon presentation of percent points, similar to the numbers your present from other studies. Then it is easier to follow your discussion.
L 312: Please, put forward that your study is performed in the Primary Healthcare by the family doctors among surgical patients, as this setting is seldom investigated for preoperative SCI, and therefore your study is among the few which have presented positive results. WHO recommends using this setting for SCI, in general (WHO manual BRIEF 2023), but they also recommend the intensive programs for preoperative SCI as shown in the reference added in the introduction chapter above. The majority of your references in the text are conducted in the hospital sector. It would further attract attention to your study results, if you could show that they are at the same level on both short and longer time as the intensive interventional RCTs conducted previously in the hospital setting (Please, look into your references 13-16).
L 325-326: This sentence is difficult to understand, please, consider rephrasing it to avoid a circular argument; it is difficult to implement as the doctors don’t do it? (but if they do it, it would already be implemented).
L 327: Please, see above
L 334-340: Please, put into focus that the participants are scheduled for surgery, thereby being at special risk for development of complications, which facilitates the motivation to quit. Please, on the other hand, also put the level addiction to nicotine measured by FTND in your study forward as a barrier for successful quitting. You mainly describe the mental aspects of being addicted.
L 347-352: This is almost repeated below in the text.
L 368-369: Please, also consider that the CO measurements are positive only for a shorter period, while cotinine covers a longer period.
L 372-379: Please, Consider if this paragraph would fit better in relation to line 327-330?
L 408-416: This paragraph is not easy to understand, as it is well established that smoking has a heavy social gradient, meaning that socially vulnerable groups have the highest frequency of smoking and a higher level of addiction
L 417-422: A bias, strengths (including the design), and limitation paragraph needs to include the bias and the strengths - and in which way these would draw the results and thereby out-balance each other to some degree. The term limitation refers to the possibility of generalization of the results to other patient groups that surgical patients, in other clinics and other countries with different cultures and so on. Please, look above for other elements to be included. As it is written in the text now, it does not present the limitations but some bias.
L 422: Please add the perspectives of this study for the patients, the clinicians including both family doctors and the surgeons, the Primary Health Care, and the society at large as well as future research in this important area.
Conclusion: The conclusion should directly answer the aim and shortly present the main results and a short recommendation to implement this program. The aim can’t be just to test if the family doctors can motivate their patients, as you have not measured the motivation levels during your trial. (e.g., Intensive smoking cessation intervention conducted by the family doctors is very effective in the perioperative period compared to advising (59% versus 12%) and in longer time (34% versus 8% or something similar).
Author contribution: It is a usual international criterion that all the authors have edited and approved the final manuscript. It may help the readers if you specify “investigation”. Does it include, recruitment of participants, the intervention, the data collection, analyzing, and interpreting the results, or have these elements been performed by more persons?
The abstract needs to be updated, accordingly.
Comments on the Quality of English Language
Included in the peer review above
Reviewer 3 Report
Comments and Suggestions for Authors
Intensive intervention on smoking cessation in patients undergoing elective surgery; the role of family physicians
Abstract
Line 27: The examinees of Group A talked to the physician five times 27 and received 120 telephone messages,
Is that the average for all sample in grp A?
Intro:
Line 43-44: Please add reference for your statement.
Line 47: The statement is not clear. Do you mean dual use of cigarette smoking and e-cigarette? where cigarette smokers use e-cig but less use?
E-cigarette use is becoming very popular nowadays, contradicting your statement.
Line 58-60: Please add reference for your statement.
The aim of the study should be mentioned clearly, so last paragraph in the introduction has to be moved up, and to clarify the aim of the study.
Intensive intervention has been mentioned many times, but there was no clear definition in that regards. Please add.
Methods:
What was the definition of smokers?
Who smokes for the last month? Three months? Years …… etc
Where any of the included participants dual users of cigarette smoke, e-cigarette or waterpipe smoke?
Line 111: Participants who did not respond to the doctor's call or gave up the study during the program or during the 12-month follow-up period were treated as smokers when presenting the results.
How many? Please add.
The principal investigator (line 135) is the same person- the family physician (line 90-91) who performed the randomization?
Please clarify.
Results:
In the abstract, it is mentioned that ‘’aged 18-56 years … ‘’. However, in the results, it is mentioned that ‘’The youngest respondent was 21 years old, and the oldest 176 was 65 years old’’
Please clarify and make it consistent throughout.
Discussion
‘’This study shows that patients preparing for elective surgery are 11 times more likely to quit smoking in the short term if their family physician applied an intensive intervention.’’
This was the conclusion of your study, but is the limited to family physician? What about smoking cessation counselling provers including nurses, GPs, physiotherapists .. etc?
This should be mentioned in the introduction and discussed in the discussion in the context of you study, and also how social support aid in smoking abstinence in addition to healthcare providers counselling.
Comments on the Quality of English LanguageMinor editing
Round 2
Reviewer 1 Report
Comments and Suggestions for Authors
Thank you for considering the recommendations.
Reviewer 3 Report
Comments and Suggestions for Authors
Authors addressed my comments